# Impact of Stem Cells on Reparative Regeneration in Abdominal and Dorsal Skin in the Rat

**DOI:** 10.3390/jdb12010006

**Published:** 2024-01-27

**Authors:** Evgeniya Kananykhina, Andrey Elchaninov, Galina Bolshakova

**Affiliations:** 1Laboratory of Growth and Development, Avtsyn Research Institute of Human Morphology of FSBI “Petrovsky National Research Centre of Surgery”, 117418 Moscow, Russia; e.kananykhina@gmail.com (E.K.); gbolshakova@gmail.com (G.B.); 2Research Institute of Molecular and Cellular Medicine, Peoples’ Friendship University of Russia, 117198 Moscow, Russia

**Keywords:** skin, regeneration, stem cells

## Abstract

A characteristic feature of repair processes in mammals is the formation of scar tissue at the site of injury, which is designed to quickly prevent contact between the internal environment of the organism and the external environment. Despite this general pattern, different organs differ in the degree of severity of scar changes in response to injury. One of the areas in which regeneration after wounding leads to the formation of a structure close to the original one is the abdominal skin of laboratory rats. Finding out the reasons for such a phenomenon is essential for the development of ways to stimulate full regeneration. The model of skin wound healing in the abdominal region of laboratory animals was reproduced in this work. It was found that the wound surface is completely epithelialized on the abdomen by 20 days, while on the back—by 30 days. The qPCR method revealed higher expression of marker genes of skin stem cells (*Sox9*, *Lgr6*, *Gli1*, *Lrig1*) in the intact skin of the abdomen compared to the back, which corresponded to a greater number of hairs with which stem cells are associated on the abdomen compared to the back. Considering that some stem cell populations are associated with hair, it can be suggested that one of the factors in faster regeneration of abdominal skin in the rat is the greater number of stem cells in this area.

## 1. Introduction

The skin is the outer covering of the body of all vertebrates. Its main functions are protective and as a barrier. In mammals, the skin consists of three layers: epidermis, dermis, and hypodermis, characterized by the presence of such skin derivatives as hair, sweat, and sebaceous glands, as well as claws, and in some species nails and horns [1].

Due to the fact that the skin is the outer covering of the body, it is often exposed to damage. Rapid wound healing is a key condition for the survival of the organism, as it prevents the penetration of infection and blood loss [2]. A characteristic feature of the wound process in mammalian skin is the formation of scar tissue in the area of injury [3]. Several stages can be distinguished in the wound process: (1) contraction of wound edges and scab formation due to platelet aggregation, which is necessary to stop bleeding and create a temporary barrier; (2) inflammatory infiltration by neutrophils, monocytes, macrophages, and lymphocytes; (3) local migration and proliferation of keratinocytes to re-epithelialize the injured skin; and (4) restoration of the underlying dermis and remodeling of its extracellular matrix [4].

The newly formed wound stroma serves as a framework for wound re-epithelialization. The sources of new keratinocyte formation are stem cells of the basal layer of the epidermis, epitheliocytes of the outlet streams of sweat and sebaceous glands, and epitheliocytes of hair follicles [5]. These cells are ascribed stem cell (SC) properties. In contrast to physiological renewal of epidermal cells, during wound healing SCs migrate from their niches to the area of injury [6,7]. Recruited progeny of SCs from any niche retain the ability to regenerate the epidermis for a long time [8,9].

Different parts of the hair follicle are known to contain SCs giving rise to different dermal derivatives [10]. The hair follicle is usually divided into three segments: the infundibulum (epithelium of the upper part of the hair follicle), the isthmus (middle part), and the lower part of the follicle (bulge zone, hair follicle). The infundibulum is the upper funnel-shaped portion that usually starts from the surface of the epidermis to the opening of the sebaceous duct and is filled with sebum. The upper part of the infundibulum has been shown to contain a population of stem cells expressing *Sca-1* (stem cell antigen 1 expressed by non-committed progenitor cells) and capable of regenerating this area [11]. The isthmus is the middle part between the sebaceous gland duct and the bulge zone. The bulge is a separate region of the outer sheath of the root sheath, the epithelial compartment, and contains stem cells capable of differentiating into hair follicle cells. In rodents, the markers of such SCs are *Gli1, MTS24, Lgr6, and Lrig1* [1,12,13]. In addition, bulge-zone keratinocytes, which are recognized as stem cells that give rise to all epithelial cells, express cytokeratin 15 [14]. The SCs of the bulge zone are more plastic, multipotent, and characterized by a more pronounced ability to proliferate compared to other populations of skin SCs [15]. Such properties of SCs of the bulge zone allow them to give rise to all types of epithelial cells of hair follicles, epidermis, sweat, and sebaceous glands [14].

The participation of different stem cell populations in skin regeneration after damage is a subject of intensive study [7]. As already mentioned, several stem cell populations are distinguished in the skin: interfollicular ones and those associated with hair follicles. Despite their diversity, the dynamics of these populations during wound healing appear to be similar. At the initial stage, there is a depletion of these stem cell populations due to their slow asymmetric division and rapid differentiation of progenitor cells. At this stage, the formation of new tissues is minimal. Further, after activation of stem cells, asymmetric division is also observed but is more active, which leads to the formation of a large number of progenitor cells and the growth of new tissues covering the wound [16]. According to other data, some populations of skin stem cells are already pre-adapted to participate in skin wound healing. For example, Lgr5+ cells need about 1 day to adapt to the new conditions of the wound process, whereas Lgr6+ cells are already ready to participate in regeneration [17].

As noted above, a characteristic feature of the wound process in mammals, including in the skin, is the formation of scar tissue. However, the severity of connective tissue scar formation varies during the healing of skin wounds of different parts of the body and also between mammalian species [18,19]. The reasons for this phenomenon are poorly understood. However, we can conclude from this that scar formation at the site of injury is not an insurmountable pattern. Considering the clinical significance of scarless healing not only of skin wounds, this issue is highly relevant. We have previously shown that a skin wound on the abdomen after full-thickness defect application closes more quickly compared to a wound of similar area in the back region [20]. At the same time, during the healing of a rat abdominal skin wound, the structure of the regenerate is close to intact skin in terms of the collagen I/III ratio, the presence of elastin, and strength characteristics, as well as the presence of skin derivatives. Although the role of stem cells in skin regeneration in mammals has been intensively studied, the contribution of stem cells to the faster and more perfect regeneration of abdominal skin compared to dorsal skin in rats has not been explored. 

Although the role of stem cells in skin regeneration in mammals has been intensively studied, the contribution of stem cells to the faster and more perfect regeneration of abdominal skin compared to dorsal skin in rats has not been explored. Thus, the aim of this work was to evaluate the expression of stem cell markers in wound healing of abdominal skin compared to dorsal skin in rats.

## 2. Materials and Methods

### 2.1. Animals

The work was performed on male Wistar rats (weight 250–300 g) obtained from the nursery Pushchino vivarium (Moscow region, Russia). All manipulations with animals were performed in accordance with the rules of work with the use of experimental animals (“International Recommendations for Conducting Biomedical Research Using Animals” of 1985, the Rules of Laboratory Practice in the Russian Federation (Order of the Ministry of Health of the Russian Federation of 91/06/2003 No. 267), and the law “On the protection of animals from abuse” chapter V, article 10, 4679-GD of 12/01/1999) and with the permission of the ethical committee of Avtsyn Research Institute of Human Morphology (protocol № 12 from 10.06.2015). 

### 2.2. Experimental Model

The experiment was performed on male Wistar rats with body weight of 250–300 g. Before skin excision, the operation site was shaved with a hair clipper and treated with antiseptic solution (chlorhexidine). In each animal, a round full-thickness skin flap (together with m. panniculus carnosus) was excised from the back (*n* = 25) or abdomen (*n* = 25). The removed skin fragment served as a control. Animals were removed from the experiment using a CO_2_ chamber on days 3, 7, 14, 20, and 30.

### 2.3. Assessment of the Wound Surface Area

The site of the injury was photographed at different periods: preoperatively, immediately after excision of the skin flap, and at 3, 7, 14, 20, and 30 days after surgery. The photos of the wound surface obtained at different times after the operation were processed in Adobe Photoshop CC program, version 14.2: the wound surface was circled along the edge; its area was measured in square millimeters.

### 2.4. Histological Study

The material for histologic study was taken on the 3rd, 7th, 14th, 20th, and 30th days after the operation. The skin flap dissected during the operation served as a control. The skin was fixed in Carnoy’s liquid, then dehydrated in three changes of 100% isopropanol and encapsulated in paraffin. Sections 5 μm thick were deparaffinized and stained by Mallory staining.

### 2.5. Immunohistochemistry

Skin flaps were frozen in liquid nitrogen immediately after collection and stored at −80 °C. Then, 5–7 μm thick cryosections were made and stained first with the antibodies to Cytokeratin15 (1:100, sc-47697, Santa Cruz Biotechnology, Dallas, TX, USA), Lrig1 (1:100, sc-514577, Santa Cruz Biotechnology, Dallas, TX, USA), Gli1 (1:100, sc-515751, Santa Cruz Biotechnology, Dallas, TX, USA), and Sox9 (1:100, ab3697, Abcam Cambridge, UK), and second with goat anti-rabbit (1:200, ab97050, Abcam, Cambridge, UK) or anti-mouse (ab6785, Abcam, Cambridge, UK) antibodies following the manufacturer’s recommendations. As a control for nonspecific binding, slides stained only with the second antibodies were used. The study was performed using a fluorescence microscope Leica DM 4000 B (Leica Microsystems, Wetzlar, Germany).

### 2.6. Proliferation Assessment

Epitheliocyte proliferation activity was studied using antibodies to Ki67 protein (1:100, ab16667, Abcam, Cambridge, UK), and the localization of the marker was visualized using a second antibody conjugated to phycoerythrin (1:200, sc-3739, Santa Cruz Biotechnology, Dallas, TX, USA). Proliferation was studied separately in the interfollicular epidermis and epitheliocytes of the hair follicle of the back and abdomen skin. The total number of epithelial cells (either in the epidermis or in the hair follicle) was counted on antibody-stained cryosections, and the number of Ki67-positive epithelial cells was counted. The proportion of Ki67+ epitheliocytes relative to the total number of epitheliocytes was determined and expressed in %.

### 2.7. Assessment of Hair Density

Skin flaps were frozen in liquid nitrogen immediately after collection and stored at −80 °C. Then, cryosections with a thickness of 5–7 μm were made. The sections were fixed with paraformaldehyde solution. Longitudinal (relative to the rat torso) skin cryosections stained with hematoxylin–eosin were photographed on a Leica DFC 295 microscope (Leica Microsystems, Wetzlar, Germany) at a total magnification of ×100. The number of hair follicles per frame was determined in the Adobe Photoshop CC program, version 14.2, and knowing the area of the frame we obtained the density of hair cover: the number of hairs per 1 mm^2^.

### 2.8. Real-Time Polymerase Chain Reaction

Skin fragments from the lesion area were placed in RNA-later buffer (QIAGEN, Venlo, Netherland), incubated for 1 day at +4 °C, and stored at −80 °C. Several skin fragments were taken to study gene expression. The skin fragment that was removed was used as a control. The area of damage was considered to be the immediate tissue inside the wound and a minimal amount of adjacent intact skin up to 1 mm wide outward from the wound edge. For each study term, material was sampled from 5 rats. Total RNA was isolated from the obtained samples using the RNeasy Fibrous tissue Mini Kit (QIAGEN, Venlo, Netherlands,). Synthesis of cDNA from the matrix of the obtained total RNA was performed using a ready-made reagent kit MMLV RT kit (Evrogen, Moscow, Russia). PCR was performed with the obtained cDNAs using ready-made qPCRmix-HS SYBR reagent kits containing fluorescent intercalating dye Sybr Green I (Evrogen, Moscow, Russia,). Primers for PCR were selected using the on-line program Primer-BLAST in accordance with generally accepted requirements. The selected primers (Table 1) were synthesized by Evrogen (Moscow, Russia). The specialized program REST 2009 (developed by M. Pfaffl and QIAGEN) was used for gene expression analysis. The housekeeping gene *Gapdh* was used as an endogenous control.

### 2.9. Statistical Analysis

Statistical analysis of the obtained data was performed using the SigmaStat 3.5 software package (version 3.5.0.54, Systat Software, Inc.). In case of normal distribution within the sample, the mean and standard deviation were determined. The data were compared using one-factor analysis of variance and the Holm–Sidak test. If the distribution differed from normal, data were presented as median and 25% and 75% quartiles (Me (Q25 ÷ Q75)), and comparisons were made using the Kruskal–Wallis rank analysis of variance and Dunn’s test at a significance level of *p* < 0.05. The percentages were compared using the z-test at a significance level of *p* < 0.05.

## 3. Results

### 3.1. Skin Wound Closure Rate

The healing process of the wound on the back occurred with the formation of a scab at the beginning and a connective tissue scar by the 30th day (Figure 1a,b). Immediately after surgery, the wound area on the back increased statistically significantly by 15.4% over the intended area (Table 2). On the 7th postoperative day, the wound area was 76.8% of the initial wound area. By the 14th day, the wound surface was completely epithelialized and its area was 17.2% of the intended defect contour. By the 20th day, a dense scab was preserved; the wound area decreased to 4.3% of the intended one. By the 30th day, the scab was practically absent, and the scar was clearly visible.

The repair of the skin wound on the abdomen was different. In the course of healing, no scab was formed; there was only a thin crust. By 20–30 days, a scar was formed at the wound site, visually almost indistinguishable from the surrounding skin (Figure 1c,d). Immediately after excision of the skin flap, the wound area on the abdomen significantly increased by 61.1% compared to the initially targeted area (Table 2). This was followed by a rapid decrease in wound area, and on the 7th postoperative day, the wound area was only 18.2% of the initial area. By the 14th day, the wound had completely closed, and the visualized scar area was 3.8% of the intended area.

### 3.2. Proliferation Activity

In regenerating skin, Ki67+ cells were identified among keratinocytes of the basal layer of the epidermis as well as in hair-associated epithelial structures (Figure 2A,B). The number of Ki67+ cells in the epidermis and hair follicle gradually decreased from 3 to 14 days of observation (Figure 2A,B). When comparing the proliferation activity, the number of Ki67+ cells was found to be higher in the interfollicular epidermis of the abdomen only 3 days after the injury; 7 and 14 days later there were no differences (Figure 2A,B).

The number of Ki67+ epitheliocytes was statistically significantly higher in the hair follicles of the abdominal skin 3, 7, and 14 days after surgery (Figure 2A,B).

### 3.3. Immunohistochemical Study of Localization of Marker Proteins of Skin SCs

Using immunohistochemical examination, we detected all the marker proteins under study in the skin. However, the localization of cells positive for one or another marker differed. CK15 was most widely expressed; CK15+ cells were found in the basal layer of the epidermis, as well as in the cells of the outer and inner epithelial sheath of the hair follicle. Numerous cells carrying Lrig1 protein were localized among the cells of the sebaceous gland; Sox9+ and Gli1+ cells were localized in the isthmus and at the hair funnel (Figure 3).

### 3.4. Hair Density

Due to the fact that SCs are associated with hair in large amounts, the easiest way to estimate the amount of SCs in the skin of the back and abdomen is to determine the density of skin hair. Hair density on the back was 2.31 (2.13 ÷ 2.81) hairs per 1 mm^2^, on the abdomen—4.38 (3.63 ÷ 5.14) hairs per 1 mm^2^—which is almost twice as much; the difference is statistically significant (Figure 4).

### 3.5. Expression of Skin Stem Cell Marker Genes

The expression levels of skin stem cell marker genes *Sox9, Gli 1, Lgr6*, and *Lrig1* were evaluated using qPCR. In intact skin, the expression level of all genes under study was statistically significantly higher in the abdominal region (*p* < 0.050) (Table 3). After skin excision, the dynamics of expression of the studied genes was as follows. In the abdomen region, the dynamics of *Sox9* and *Lgr6* gene expression coincide: expression decreased at 3, 7, and 20 days after surgery; at the other terms, the changes were insignificant (Figure 5). *Gli1* gene expression increases by the 20th day after surgery and remains at the level of intact skin during the rest periods. *Lrig1* gene expression decreases by 20 days after surgery and remains at the level of intact skin during the rest of the period. In the dorsal skin, *Sox9* gene expression slightly decreases on the 7th day after surgery. The expression of *Lgr6* and *Gli1* genes decreases by 7 days and remains at the level of intact skin during the rest periods. *Lrig1* gene expression increases (11-fold) by the 30th day and remains at the level of intact skin during the rest of the period.

## 4. Discussion

The formation of scar tissue in response to damage is a characteristic feature of the course of reparative processes in mammals [1]. The formation of scar tissue leads to a rapid restoration of the lost integrity of certain structures of the organism. However, despite the obvious advantages of this method of repair, it has significant negative sides. The scar by its functional characteristics compares unfavorably to the native skin due to its mechanical characteristics. The probability of rupture at the defect site increases, not to mention the aesthetic component. Thus, the problem of scarless healing is relevant for the most diverse branches of medicine.

Despite the universality of the mechanism of repair through scar formation in mammals, the degree of its expression varies in different organs [19,21]. This also applies to the skin. A practically scarless healing of a full-thickness abdominal skin wound in a rat was described earlier [18]. The study of molecular and cellular mechanisms of such a different reaction within one organ seems to be extremely relevant for the creation of means stimulating full regeneration.

We have shown that healing of the abdominal skin wound in the rat leads to the formation of a more complete regenerate, the structure of which is maximally close to the intact skin. This is manifested by a higher ratio of more mature type I collagen to type III collagen, the presence of elastin, and strength characteristics [20]. The reasons for the formation of such a regenerate in the area of abdominal skin wounds, apparently, lie in the peculiarities of the course of the inflammatory process. It was found that on the 7th day after surgery in the wound on the abdomen there were statistically significantly fewer CD68-positive cells (macrophages) and more FAP-α-positive fibroblasts, indicating an earlier transition to the onset of collagenogenesis [20].

In addition to the fact that the formed regenerate in the area of the abdominal skin wound is closer in structure to the intact skin, the rate of epithelialization and wound closure itself is higher in the abdominal area. This dynamics of the wound process is determined by a number of reasons. First of all, as we have shown earlier, the abdominal wound closes more quickly due to the pronounced insertion growth of the skin [22]. The data on pronounced insertion growth are in agreement with the results of the study of proliferation activity during healing of back and abdominal skin presented in this article. The number of proliferating Ki67+ cells was significantly higher in keratinocytes of the interfollicular epidermis and epitheliocytes of the hair follicle. The second reason is a faster contraction of the wound edges in the abdominal region. This can be clearly seen from the data presented in Table 2. Immediately after application, the wound area on the abdomen increases sharply, but on the 3rd day it also shrinks sharply. However, on the 7th day and later, the dynamics of faster wound closure in the abdomen area remain, which cannot be explained by contraction alone. Here, the insertion growth makes a greater contribution. 

Earlier, we studied the thickness of the periwound zone and its area with the help of ink marks. A significant increase in the area of the periwound zone was shown starting at 14 days. By 30 days, the periwound zone on the abdomen was almost two times larger than the similarly marked area before the defect application. At the same time, the skin thickness in this zone did not differ from the intact skin [22]. This all points to cell proliferation in the periwound zone, not in early terms, but in the phase of remodeling. The reason for the more pronounced insertion growth in the healing of abdominal skin wounds compared to dorsal skin wounds is yet to be clarified. It is probably due to the peculiarities of the structure of the abdominal skin compared to the back skin. One such feature may be a greater number of stem cells in the abdominal skin of rats. This is indirectly indicated by the greater density of hair in the abdominal skin compared to the dorsal skin, since several populations of skin stem cells are associated with hair. In addition, less pronounced inflammation may lead to faster closure of the wound surface in the abdominal skin, as indicated by a significant decrease in the number of CD68+ cells in the area of the abdominal skin wound on the 7th day after surgery [20].

In the present study, we show that the abdominal skin has a significantly higher hair density compared to the dorsum. Contradictory data regarding the density of hair on the back and abdomen of rats have been found in the scientific literature. According to some data, the density of hair on the back exceeds that on the abdomen, which contradicts our data [18]. However, in this work, only visual assessment was performed without using quantitative methods of analysis.

Several subpopulations of epithelial stem cells are known to be described in the skin, most of which are associated with hair and sebaceous glands [1]. Thus, higher hair density leads to higher stem cell density in the abdominal skin, which underlies the faster closure of the wound surface in the abdominal skin region.

The higher stem cell abundance in the abdominal skin is also indicated by the higher expression of the marker genes *Sox9, Lgr6, Gli1*, and *Lrig1* in this region. However, after wounding, the expression of the above genes specifically in the wounded area of the abdominal skin appears to be significantly reduced compared to the intact skin of this region for a long period. This phenomenon can be explained by the fact that we compared the expression level of marker genes in the wound with the intact skin of this region. Due to the higher density of hair in this region, their disappearance upon wounding results in a marked drop in their expression level. Epithelialization of the wound does not lead to restoration of the expression level of marker genes as happens in the back wound area. This is due to the fact that if hair is formed here occasionally, its amount is much inferior to that in the intact skin.

It should be noted that we evaluated the expression of a particular marker gene relative to the intact skin of the same area (the removed flap served as a control). In addition, the expression of one or another marker gene was studied in the wound area, which lacks any epithelial lining at early stages. On this basis, a sharp drop in the expression of the studied genes immediately after excision of a full-layer skin flap and a sharp increase in the expression of the above genes after epithelialization is completed and, probably, the number of all populations of stem cells associated with epidermis and hair being restored is quite understandable. Thus, the demonstrated dynamics of marker gene expression reflect the process of completion of epithelialization and restoration of skin integrity. In addition, given that we have previously shown a pronounced contribution of skin insertion growth in wound closure, it is likely that the activity of skin stem cells changed primarily in the intact skin surrounding the wound. This is consistent with the previously obtained data [20].

On the other hand, during wound healing there may be no increase in the number of stem cells, rather even the contrary—depletion on the background of increased proliferation [16], which is also consistent with our data, according to which there is a sharp decrease in the expression of stem cell marker genes after skin damage [22,23].

Thus, the repair of abdominal skin in rats is faster compared to back skin injury. This phenomenon is based on the difference in the structure of dorsal and abdominal skin. In the present study, we have shown a greater density of hair in the abdominal skin, which is associated with a higher level of expression of the skin stem cell marker genes *Sox9*, *Lgr6*, *Gli1*, and *Lrig1*.

## Figures and Tables

**Figure 1 jdb-12-00006-f001:**
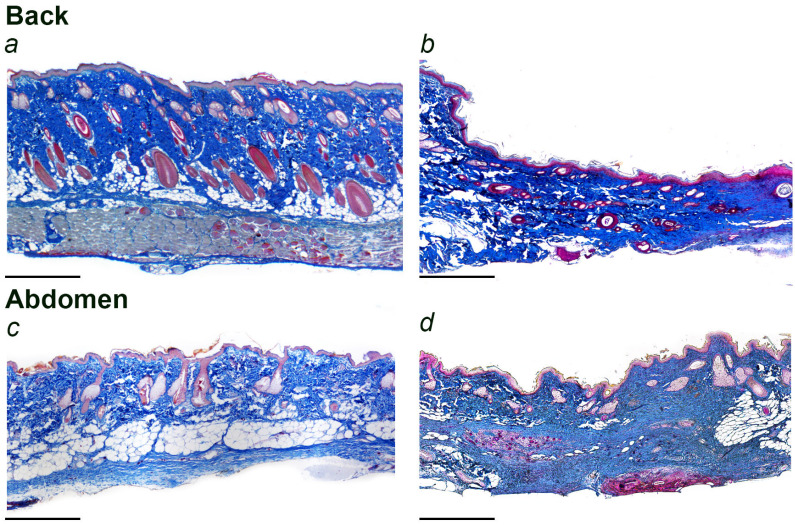
The skin of the back and abdomen of the rat before surgery and the site of the skin wound 30 days after surgery. (**a**)—back skin before surgery, (**b**)—site of the skin wound on the back 30 days after surgery, (**c**)—abdominal skin before surgery, (**d**)—site of the skin wound on the abdomen 30 days after surgery. Scale section 500 µm. Mallory staining.

**Figure 2 jdb-12-00006-f002:**
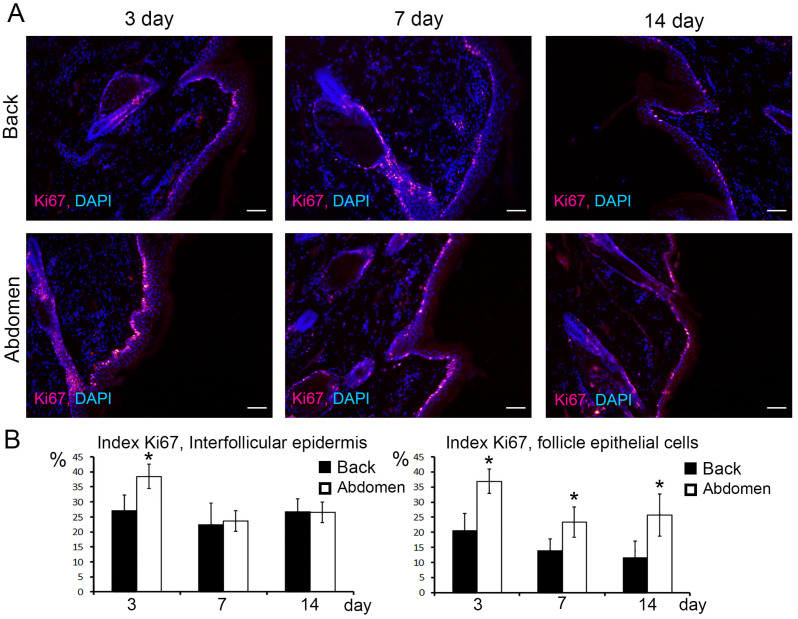
Immunohistochemical study of proliferative activity. (**A**) Detection of Ki67 proliferation marker in keratinocytes of intrafollicular epidermis and epitheliocytes of hair follicle in back and abdomen skin (red glow), nuclei stained with DAPI (blue glow), scale bar—50 μm. (**B**) Ki67 index in % in keratinocytes of intrafollicular epidermis and hair follicle epitheliocytes in back and abdominal skin. Standard deviation was used as error bars, *—statistically significant differences compared to the corresponding index at the given term in the back skin (*p* < 0.05).

**Figure 3 jdb-12-00006-f003:**
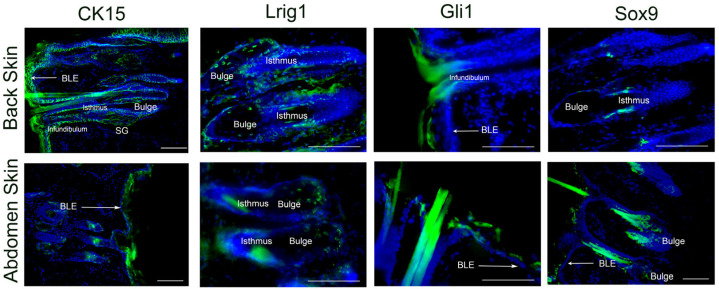
Immunohistochemical staining of intact skin of the back and abdomen with antibodies to skin stem cells markers: CK15-cytokeratin15, Lrig1, Gli1, Sox9 (green glow). Cell nuclei are stained in blue (DAPI dye), specific marker in green. The scale marker is 100 μm, SG—sebaceous gland, BLE—basal layer of the epidermis.

**Figure 4 jdb-12-00006-f004:**
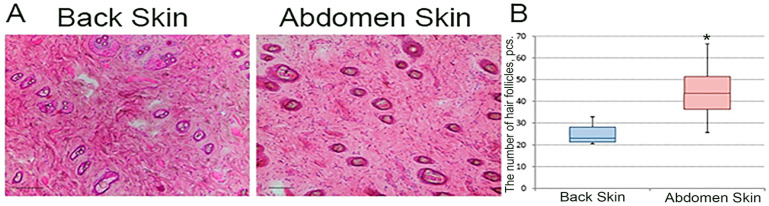
Longitudinal skin sections stained with hematoxylin and eosin, scale bar 100 μm (**A**). Number of hairs on the skin of the back and abdomen per square mm, * denotes statistically significant differences between groups, *p* < 0.001 (**B**).

**Figure 5 jdb-12-00006-f005:**
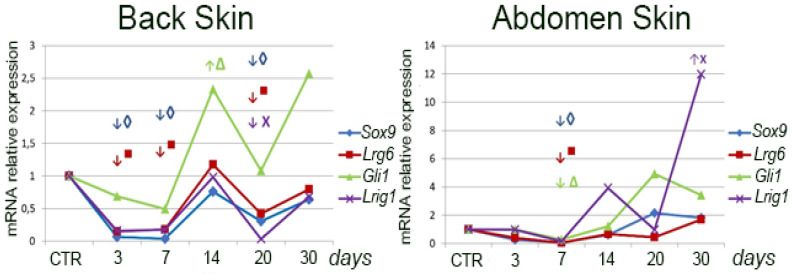
Changes in the expression of SCs genes of back and abdominal skin in the process of wound healing. Different genes are highlighted in color; statistically significant differences (compared to the control) in expression are indicated by arrows: ↑—increase, ↓—decrease.

**Table 1 jdb-12-00006-t001:** Sequence of primers.

Gene	Forward	Reverse
*Gapdh*	gcgagatcccgctaacatca	cccttccacgatgccaaagt
*Sox9*	agaggccaccgaacagac	tgctcagctcaccgatgtc
*Lgr6*	aggatggcatcatgctgtca	ccgtgaggttgttcatactg
*Gli1*	caaggcccagtacatgctg	tcctattctggtgcttggca
*Lrig1*	gaggacctggggtagtaggc	aaccatattgcaggcgctca

**Table 2 jdb-12-00006-t002:** Area of full-thickness wound and periwound area on the abdomen and back of rats at different terms after surgery (mean ± standard deviation).

Time after Skin Excision	Back	Abdomen
	Wound Area, mm^2^	Wound Area, mm^2^
**Targeted wound area**	189.2 ± 13.6 *#	165.9 ± 13.3 *#
**Wound area, immediately after surgery**	223.8 ± 13.6 *#	271.4 ± 41.9 *#
3	195.8 ± 27.6 *#	104.8 ± 36.3 *#
7	145.4 ± 20.7 *#	30.2 ± 14.9 *#
14	32.6 ± 13.0 *#	6.4 ± 3.9 *#
20	8.1 ± 5.0 *#	0 *#
30	1.1 ± 1.5 *#	0 #

*—differences at different postoperative time points within the same area (back or abdomen), #—differences between back and abdomen at the same time point.

**Table 3 jdb-12-00006-t003:** Relative expression level of stem cell marker genes in intact skin (Me—median, Q1—first quartile, Q3—third quartile).

Gene	Back Skin			Abdomen Skin		
	Me	Q1	Q3	Me	Q1	Q3
*Sox9*	**0.005**	0.004	0.018	**0.037**	0.013	0.054
*Lgr6*	**0.73**	0.391	2.801	**2.6**	1.353	6.333
*Gli1*	**0.43**	0.284	1.21	**1.44**	0.963	2.345
*Lrig1*	**0.01**	0.008	0.06	**0.09**	0.029	0.352

## Data Availability

All data generated or analyzed during this study are included in this published article. Additional data will be made available on request.

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
