# Peer review of "Impact of Stem Cells on Reparative Regeneration in Abdominal and Dorsal Skin in the Rat"

_jdb, 2024, doi:10.3390/jdb12010006_

Round 1

Reviewer 1 Report

Comments and Suggestions for Authors

General comments

This paper includes significant findings on skin regeneration in mammals. However, there are an overstatement, other major issue, and minor ones. The details are as follows.

Major comments

L21: “Thus, it can be concluded that the faster healing of full-thickness skin wound on the abdomen is due to a greater number of poorly differentiated cells in this area.” is an overstatement. It should be proven by experiments of increasing and decreasing of “poorly differentiated cells”. I recommend the authors to use “Thus, it is suggested that…”, “We propose the hypothesis that…”, etc.

L132: “the lesion area” is ambiguous to identify what the samples are. Clarify their outlines. Otherwise, it is much difficult to discuss Figure 4.

Minor comments

L51: “isthmus (isthmus, middle part) and the lower part of the follicle (bulge zone, hair follicle)” would be a mistake for “isthmus (the middle part) and bulge zone (the lower part)”.

L60: Gli1 is not mentioned in Ref. 12 or 13.

L182: ”2” of  “mm2” should be superscript. There are the same descriptions in L199 and L200.

L182: It’s better to align the commas vertically.

L187: Explain here why CK15 was chosen and cite a paper that demonstrates that CK15 is a marker for skin stem cells.

L191: Use symbols such as arrows to indicate the location of the locations of the basal layer of the epidermis, the outer and inner epithelial sheath of the hair follicle, the sebaceous gland, the isthmus, and the hair funnel.

L192: Green signals for Gli1 and Sox9 seem to be located in abdomen skin although they are not located in hairs in back skin. It is difficult to understand that.

L196: According to the Reference 16, the skin on the ventral side has less hair, which contradicts the results of this study. It is better to mention this in the manuscript although it might not be essential.

L221: Explanations of what Me, 1Q, and 3Q are are required.

L221: Unify the significant digits or unify the horizontal positions of the decimal points.

L280: This "Conclusion" is just a simple description of the main findings. Eliminate this section or include the main discussion in it.

Author Response

Dear editor and reviewers!

Thank you for your in-depth analysis of our work and your comments and remarks! Below we have tried to answer them consistently! The text of the manuscript has been modified as recommended. All changes are highlighted in yellow.

General comments

This paper includes significant findings on skin regeneration in mammals. However, there are an overstatement, other major issue, and minor ones. The details are as follows.

Major comments

L21: “Thus, it can be concluded that the faster healing of full-thickness skin wound on the abdomen is due to a greater number of poorly differentiated cells in this area.” is an overstatement. It should be proven by experiments of increasing and decreasing of “poorly differentiated cells”. I recommend the authors to use “Thus, it is suggested that…”, “We propose the hypothesis that…”, etc.

Thank you for your valuable observation. Indeed, many cellular components take part in wound healing. At the same time, we have shown in earlier works that the skin of the back and abdomen in rats differs in its structure: the expression of dermis layers, thickness of subcutaneous tissue. In the present study, it was found that another distinctive feature of the skin of the abdomen compared to the back is the greater number of hair follicles. Considering that some stem cell populations are associated with hair, it can be suggested that one of the factors for faster regeneration of abdominal skin in the rat is the greater number of stem cells in this area.

L132: “the lesion area” is ambiguous to identify what the samples are. Clarify their outlines. Otherwise, it is much difficult to discuss Figure 4.

Thank you for the important question! Several skin fragments were taken to study gene expression. The skin fragment that was removed was used as a control. The area of damage was considered to be the immediate tissue inside the wound and a minimal amount of adjacent intact skin up to 1 mm wide outward from the wound edge.

Minor comments

L51: “isthmus (isthmus, middle part) and the lower part of the follicle (bulge zone, hair follicle)” would be a mistake for “isthmus (the middle part) and bulge zone (the lower part)”.

Thanks for the comment, the text has been corrected.

L60: Gli1 is not mentioned in Ref. 12 or 13.

Thanks for the comment, the text has been corrected.

Takeo M, Lee W, Ito M. Wound healing and skin regeneration. Cold Spring Harb Perspect Med. 2015 Jan 5;5(1):a023267. doi: 10.1101/cshperspect.a023267.

L182: ”2” of  “mm2” should be superscript. There are the same descriptions in L199 and L200.

Thanks for the comment, the text has been corrected.

L182: It’s better to align the commas vertically.

Thanks for the comment, the text has been corrected.

L187: Explain here why CK15 was chosen and cite a paper that demonstrates that CK15 is a marker for skin stem cells.

Thank you for your valuable comment and the text has been edited accordingly. Antibodies to cytokeratin 15 stain bulge-zone keratinocytes, which are recognized as stem cells that give rise to all epithelial cells (Morris RJ, Liu Y, Marles L, Yang Z, Trempus C, Li S, Lin JS, Sawicki JA, Cotsarelis G. Capturing and profiling adult hair follicle stem cells. Nat Biotechnol. 2004 Apr;22(4):411-7. doi: 10.1038/nbt950. Epub 2004 Mar 14.)

L191: Use symbols such as arrows to indicate the location of the locations of the basal layer of the epidermis, the outer and inner epithelial sheath of the hair follicle, the    , the isthmus, and the hair funnel.

Thank you for your valuable comment In the microphotographs, where possible, we indicated the components of hair and skin.

L192: Green signals for Gli1 and Sox9 seem to be located in abdomen skin although they are not located in hairs in back skin. It is difficult to understand that.

Thank you for your valuable comment. Probably the signal from staining in the back area is somewhat weaker compared to the abdominal skin. Marked cells positive for one or another marker with arrows.

L196: According to the Reference 16, the skin on the ventral side has less hair, which contradicts the results of this study. It is better to mention this in the manuscript although it might not be essential.

Thank you for your valuable comment. Yes, the author mentioned that there is less hair on the abdomen of rats, but, unfortunately, no numerical data and method of counting are presented, so we did not quote this aspect. Visually it seems that the hair cover is denser on the back, but histological counting on cryosections (to exclude the factor of tissue deformation during formalin wiring) showed that it is not so.

L221: Explanations of what Me, 1Q, and 3Q are required.

Thank you for your comment, the table has been expanded.  Me - median, Q1 - first quartile, Q3 - third quartile.

L221: Unify the significant digits or unify the horizontal positions of the decimal points.

Thanks for the comment, the text has been corrected.

L280: This "Conclusion" is just a simple description of the main findings. Eliminate this section or include the main discussion in it.

Thanks for the comment, the text has been corrected.

Yours faithfully,

Dr. Andrey Elchaninov

Head of Laboratory of Growth and Development,

Laboratory of Growth and Development, Avtsyn Research Institute of Human Morphology of FSBI "Petrovsky National Research Centre of Surgery", Moscow, Russia,

3 Tsyurupa Street, Moscow, Russia. E-mail: [email protected]; phone: +7 (916) 8885292

Reviewer 2 Report

Comments and Suggestions for Authors

The researchers have approached the interesting question of why wound healing is faster in the abdominal vs. dorsal skin of the rat. The finding of increased HFs in ventral skin and increased SC markers is interesting and may or may not have a role in the faster wound healing seen in the abdominal skin. However, no connection is shown between increased SC markers and improved healing. In fact, the rate of wound healing appears to be dramatically greater for only the first 3 days, and the authors should decipher what is different in abdominal skin healing during this time period.

In summation, the authors have previously shown that ventral wounds heal faster. Here they show that hair density is greater on the ventral surface. In keeping, they also show that stem cell genes are more highly expressed in the abdominal skin. During abdominal healing Gli1 is increased at day 20 (fig. 4) (I think “14th day” was a typo). In dorsal skin Lrig 1 increases at day 30.

Major issues:

This study does nothing to show that the faster regeneration of abdominal skin is due to higher stem cell activity. The title must be changed. Finding increased stem cell markers may indicate a higher number of stem cells or just increased expression of the marker. It does not indicate higher activity.

Essentially,  substantial new experimental data is needed to show some relevance for stem cell or other changes in the faster wound healing in the first 3 days. 

After the initial remarkable decrease in abdominal wound area at day 3, a quick plot of wound area vs time reveals a lesser slope (i.e. the rate of wound healing is slower) in the abdominal skin after the first 3 days. Whether it is significantly slower could be determined by the authors. Since the maximal post-wounding proliferation response in rodents is around day 4, the early difference is suggestive of greater wound contraction, or at least other factors than stem cell proliferation. This initial increased rate of wound healing is incredibly interesting, and the authors should focus on changes in this period of rapid wound shrinkage. Is there already increased proliferation at this point in abdominal skin but not back skin?

The increase in Gli1 (abdomen day 20) and Lrig1 (back day 30) are each occurring when the wound has just healed. This is not particularly supportive of a role in the faster abdominal wound healing. There is no discussion of the relationship of Fig. 4 to the hypothesis.

The introduction does not include established knowledge regarding stem cells and wound healing and should include knowledge about stem cells and healing from Aragona et al, 2017 and others.

Table 3: What is the sample size for the qPCR? What is Me short for – Mean? Are the values significantly different?

Smaller issues.

Ventral hair has a greater proportion of zig-zag hairs and a shorter anagen. This is not discussed, Do these relate to the alterations in wound healing?

Author Response

Dear editor and reviewers!

Thank you for your in-depth analysis of our work and your comments and remarks! Below we have tried to answer them consistently! The text of the manuscript has been modified as recommended. All changes are highlighted in yellow.

The researchers have approached the interesting question of why wound healing is faster in the abdominal vs. dorsal skin of the rat. The finding of increased HFs in ventral skin and increased SC markers is interesting and may or may not have a role in the faster wound healing seen in the abdominal skin. However, no connection is shown between increased SC markers and improved healing. In fact, the rate of wound healing appears to be dramatically greater for only the first 3 days, and the authors should decipher what is different in abdominal skin healing during this time period.

We have previously shown that wound healing on the abdomen is faster. This dynamics of the wound process is determined by a number of reasons. First of all, as we have shown earlier, the abdominal wound closes faster due to the pronounced insertion growth of the skin. The second reason is a faster contraction of the wound edges in the abdominal region. This can be clearly seen from the data presented in Table 2. Immediately after application, the wound area on the abdomen increases sharply, but on the 3rd day it also shrinks sharply. However, on the 7th day and later, the dynamics of faster wound closure in the abdomen area remains, which cannot be explained by contraction alone. Here the insertion growth makes a greater contribution.

Earlier we have studied the thickness of the periwound zone and its area with the help of ink marks. A significant increase in the area of the peri-wound zone was shown starting at 14 days. By 30 days, the periwound zone on the abdomen is almost 2 times larger than the similarly marked area before the defect application. At the same time, the skin thickness in this zone does not differ from the intact skin. This all points to cell proliferation in the periwound zone, but not in early terms, but in the phase of remodeling.

What is the reason for the more pronounced insertion growth in the healing of abdominal skin wounds compared to dorsal skin wounds is yet to be clarified. It is probably due to the peculiarities of the structure of the abdominal skin compared to the back skin. One of such features may be a greater number of stem cells in the abdominal skin of rats. This is indirectly indicated by the greater density of hair in the abdominal skin compared to the dorsal skin, since several populations of skin stem cells are associated with hair. Also, we found higher cell proliferation activity as assessed by the proliferation marker Ki67. This was true for both interfollicular epidermis and hair follicle epitheliocytes (These data are included in the revised version of the manuscript). In addition, less pronounced inflammation may lead to faster closure of the wound surface in the abdominal skin, as indicated by a significant decrease in the number of CD68+ cells in the area of the abdominal skin wound on the 7th day after surgery.

In summation, the authors have previously shown that ventral wounds heal faster. Here they show that hair density is greater on the ventral surface. In keeping, they also show that stem cell genes are more highly expressed in the abdominal skin. During abdominal healing Gli1 is increased at day 20 (fig. 4) (I think “14th day” was a typo). In dorsal skin Lrig 1 increases at day 30.

Thanks for the comment, the text has been corrected.

Major issues:

This study does nothing to show that the faster regeneration of abdominal skin is due to higher stem cell activity. The title must be changed. Finding increased stem cell markers may indicate a higher number of stem cells or just increased expression of the marker. It does not indicate higher activity.

Thank you for your comment, we completely agree with you. The title has been corrected to a more correct one.

Essentially,  substantial new experimental data is needed to show some relevance for stem cell or other changes in the faster wound healing in the first 3 days.

After the initial remarkable decrease in abdominal wound area at day 3, a quick plot of wound area vs time reveals a lesser slope (i.e. the rate of wound healing is slower) in the abdominal skin after the first 3 days. Whether it is significantly slower could be determined by the authors. Since the maximal post-wounding proliferation response in rodents is around day 4, the early difference is suggestive of greater wound contraction, or at least other factors than stem cell proliferation. This initial increased rate of wound healing is incredibly interesting, and the authors should focus on changes in this period of rapid wound shrinkage. Is there already increased proliferation at this point in abdominal skin but not back skin?

What is the reason for the more pronounced insertion growth in the healing of abdominal skin wounds compared to dorsal skin wounds is yet to be clarified. It is probably due to the peculiarities of the structure of the abdominal skin compared to the back skin. One of such features may be a greater number of stem cells in the abdominal skin of rats. This is indirectly indicated by the greater density of hair in the abdominal skin compared to the dorsal skin, since several populations of stem cells skin cells are associated with hair. In addition, we found higher cell proliferation activity as assessed by the proliferation marker Ki67. This was true for both interfollicular epidermis and hair follicle epitheliocytes. These data are included in the revised version of the manuscript. Also, less pronounced inflammation may lead to faster closure of the wound surface in the abdominal skin, as indicated by a significant decrease in the number of CD68+ cells in the area of the abdominal skin wound on the 7th day after surgery.

The increase in Gli1 (abdomen day 20) and Lrig1 (back day 30) are each occurring when the wound has just healed. This is not particularly supportive of a role in the faster abdominal wound healing. There is no discussion of the relationship of Fig. 4 to the hypothesis.

Thank you for the important question!  In the section discussing the results, we gave some thoughts on this subject (L260-277). It should be noted that we evaluated the expression of a particular marker gene relative to the intact skin of the same area (the removed flap served as a control). In addition, the expression of one or another marker gene was studied in the wound area, which lacks any epithelial lining at early stages. On this basis, a sharp drop in the expression of the studied genes immediately after excision of a full-layer skin flap and a sharp increase in the expression of the above genes after epithelization is completed and, probably, the number of all populations of stem cells associated with epidermis and hair is restored are quite understandable. Thus, the demonstrated dynamics of marker gene expression reflects the process of completion of epithelialization and restoration of skin integrity.  In addition, given that we have previously shown a pronounced contribution of skin insertion growth in wound closure, it is likely that the activity of skin stem cells changed primarily in the intact skin surrounding the wound.

The introduction does not include established knowledge regarding stem cells and wound healing and should include knowledge about stem cells and healing from Aragona et al, 2017 and others.

Thank you for the important observation. Information on the role of stem cells in skin regeneration is included in the Introduction section of the article.

Table 3: What is the sample size for the qPCR? What is Me short for – Mean? Are the values significantly different?

For PCR we took 5 rats for each term. For all experimental samples, PCR was performed on control skin flaps (a piece of skin flap was saved when cutting out the skin flap).

Smaller issues.

Ventral hair has a greater proportion of zig-zag hairs and a shorter anagen. This is not discussed, Do these relate to the alterations in wound healing?

This is a very interesting aspect that unfortunately we were not aware of. Probably because all the results on hair typing have been obtained on mice, and we have a study on rats. Even though they are rodents, in order to compare you have to show that rats have it as well. This is likely to be confirmed. It would be interesting to investigate that. Probably, for new hair to appear in the regenerate it is very important that the hair stem cells are "in the right place, at the right time" and maybe just a short anagen promotes this.

Yours faithfully,

Dr. Andrey Elchaninov

Head of Laboratory of Growth and Development,

Laboratory of Growth and Development, Avtsyn Research Institute of Human Morphology of FSBI "Petrovsky National Research Centre of Surgery", Moscow, Russia,

3 Tsyurupa Street, Moscow, Russia. E-mail: [email protected]; phone: +7 (916) 8885292
